# On the Origin and Evolution of *Drosophila* New Genes during Spermatogenesis

**DOI:** 10.3390/genes12111796

**Published:** 2021-11-15

**Authors:** Qianwei Su, Huangyi He, Qi Zhou

**Affiliations:** 1The MOE Key Laboratory of Biosystems Homeostasis & Protection and Zhejiang Provincial Key Laboratory for Cancer Molecular Cell Biology, Life Sciences Institute, Zhejiang University, Hangzhou 310058, China; 12007115@zju.edu.cn (Q.S.); hehuangyi@zju.edu.cn (H.H.); 2Department of Neuroscience and Developmental Biology, University of Vienna, 1030 Vienna, Austria; 3Center for Reproductive Medicine, The 2nd Affiliated Hospital, School of Medicine, Zhejiang University, Hangzhou 310052, China

**Keywords:** new genes, single-cell RNA-seq, meiotic sex chromosome inactivation, out of the X

## Abstract

The origin of functional new genes is a basic biological process that has significant contribution to organismal diversity. Previous studies in both *Drosophila* and mammals showed that new genes tend to be expressed in testes and avoid the X chromosome, presumably because of meiotic sex chromosome inactivation (MSCI). Here, we analyze the published single-cell transcriptome data of *Drosophila* adult testis and find an enrichment of male germline mitotic genes, but an underrepresentation of meiotic genes on the X chromosome. This can be attributed to an excess of autosomal meiotic genes that were derived from their X-linked mitotic progenitors, which provides direct cell-level evidence for MSCI in *Drosophila*. We reveal that new genes, particularly those produced by retrotransposition, tend to exhibit an expression shift toward late spermatogenesis compared with their parental copies, probably due to the more intensive sperm competition or sexual conflict. Our results dissect the complex factors including age, the origination mechanisms and the chromosomal locations that influence the new gene origination and evolution in testes, and identify new gene cases that show divergent cell-level expression patterns from their progenitors for future functional studies.

## 1. Introduction

The origin of new genes is one of the fundamental processes underlying the production of functional novelties and organismal diversity [1,2]. This process is inevitably shaped by pre-existing regulatory differences between organs of the individual [3], and those between autosomes and sex chromosomes in the genome [4]. Extensive case and large genomic studies in *Drosophila* and mammals discovered two surprisingly convergent patterns between the two distantly related phyla. First, testes, compared with other organs, seem to play a more important role in permitting the nascent new genes to originate, and to subsequently adopt novel functions in other tissues by evolutionary time (the ‘out of the testis’ hypothesis) [5]. New genes, especially those of young ages (e.g., species-specific new genes) are more likely to have a testis-biased expression pattern than the other ‘older’ new genes (new genes that are shared by multiple species) or single-copy genes [6,7,8]. This is probably because intensive sperm competition and sexual conflict drive testes to evolve overall a more rapidly evolving transcriptomic environment than other organs that may accommodate new genes. In addition, during mammalian spermatogenesis, meiotic and postmeiotic cells (spermatocytes and spermatids) were reported to confer a more permissive open chromatin state that may allow and expose more promiscuous expression of nascent new gene copies to natural selection [9,10]. These new gene copies are either to be purged by natural selection, or to be fixed within the population if they confer novel beneficial functions. 

A second general pattern regarding the birth of new genes is that the X chromosome seems to be producing an excess of testis-biased new genes that are relocated onto the autosomes in both *Drosophila* and mammals [11,12]. This ‘out of the X’ pattern has been found for the new genes generated by retrotransposition or DNA-based duplications [7], suggesting that this is not impacted by the mechanism of how new genes were generated, but by the complex different molecular and evolutionary factors acting on the X chromosome vs. autosomes [4]. Such factors include meiotic sex chromosome inactivation (MSCI), a process that was well documented in mammals with both X and Y chromosomes forming a heterochromatic ‘XY body’ in spermatocytes [13]. More specifically, the ‘out of the X’ pattern for retrogenes was found to be only started after eutherian lineages were split, which was taken as evidence for the origin of MSCI in the eutherian ancestor [14]. Similar patterns of precocious silencing of X-linked genes in spermatocytes have also been reported in *C. elegans* [15] and *Drosophila* [16] by cytogenetic or transgenic studies. A second factor is the sexually antagonistic selection that is also expected to drive nascent male-biased genes off the female-biasedly transmitted X chromosome [17]. These factors, together with the lack of canonical dosage compensation of X chromosome in the male germline [18], lead to an underrepresentation of male-biased genes on the X chromosome, i.e., demasculinization of the X [19]. 

Previous studies of new genes used the whole-testis RNA-seq [19,20], or different manually dissected parts of *Drosophila* testis that are enriched for different stages of cells in spermatogenesis for microarray [21]. Direct cell-level evidence for MSCI driving the new gene out of the X chromosome is still lacking. In addition, debates about whether there is MSCI in *Drosophila* [22,23] have lasted until very recently. A new single-cell RNA-seq (scRNA-seq) study of larvae testes of *D. melanogaster* provided novel cell-level expression evidence and resolution of distinctive nuclear features of X chromosome vs. autosomes that supported the existence of MSCI of some but not all X-linked genes [24]. This study also uncovered that the gene-poor dot chromosome, a former X chromosome in non-Drosophila Diptera species [25], exhibited a similar dynamic expression pattern with that of X chromosome during spermatogenesis, probably due to their close nuclear territory that subjects them to the same regulatory program. Another scRNA-seq of adult testes found that de novo genes, which originated from non-coding DNA sequences, showed a distinct cell type distribution across expressed testis cell types between segregating vs. fixed de novo genes [26]. This work revealed a dynamic cell-level expression pattern of X-linked genes vs. autosomal genes, or that of de novo genes vs. other genes in the testes; and also offered a great resource to re-examine several hypotheses regarding the new gene evolution in testes which previously cannot be precisely addressed due to the relatively lower resolution of bulk RNA-seq or microarray data. Here, we reanalyze the testis scRNA-seq data, with the focus on comparing the expressed cell types or stages during spermatogenesis between the new gene and its parental gene. Our aim is to reveal how the age, origination mechanisms, and chromosome location impact the evolution of new genes across different testis cell types. Specifically, if MSCI exists in *Drosophila* and drives the relocation of X-borne new genes onto the autosomes, we would expect that a shift of autosomal new genes to be expressed toward the middle or late spermatogenesis (e.g., in spermatocytes or spermatids after spermatogenesis enters meiosis) stages, relative to their parental genes that are expressed in the early spermatogenesis (mitotic) stage (e.g., spermatogonia). Test of this hypothesis is to provide mounting evidence for MSCI in *Drosophila*, and provide more insights into the molecular driving forces of new gene evolution in testes. 

## 2. Results

### 2.1. Comparing the Expression Patterns of Genes between Chromosomes during Spermatogenesis

In order to compare the new gene and parental gene, and test the aforementioned hypothesis about MSCI, we first prepare two datasets: one is the identified new genes and their parental genes among different *Drosophila* lineages from GenTree (http://gentree.ioz.ac.cn/, accessed on 1 March 2021) [27], with the information of age and origination mechanisms of new genes’ annotated (Figure 1a). In general, new genes tend to show a shorter gene length, and less exons (*p* < 0.05, Wilcoxon rank sum test) compared with their parental genes (Appendix A), because many new genes were produced by retrotransposition without any introns (hence a shorter gene length), or partial gene duplications. For the chromosomal location, younger new genes (branch 5,6) are enriched (*p* < 0.05, Fisher’s exact test) on the X chromosome, whereas older new genes (branch1) are enriched on autosomes, which is consistent with the reported result [6]. There does not seem to be a significant difference in distance to nearby meiotic recombination hotspots [28], nor the distance to the nearby transposable elements between the new genes and the genomic background (Appendix A). The new gene’s age is inferred based on the presence or absence of the focal new gene in *D. melanogaster* or multiple *Drosophila* species including *D. melanogaster*, i.e., in the ancestor of all these species. Additionally, the origination mechanisms were divided into DNA-based duplication, RNA-based duplication (retrotransposition), or de novo evolution [1,29], based on their possibly different patterns emerging from later expression comparison between parental vs. new genes. For example, new genes produced by retrotransposition or located on a different chromosome compared with the parental genes are more likely to share different *cis*-regulatory elements with those of their parental genes and evolved novel expression patterns. In particular, de novo genes have been recently analyzed with the *Drosophila* adult testis scRNA-seq dataset and are not the focus of this study, nor they help to clarify the role of MSCI during new gene evolution [26]. A second dataset is that we use the scRNA-seq of adult testes of *D. melanogaster* and annotate every gene for their major expressed stages during spermatogenesis. We use the adult testis data because some new or parental genes in the following analyses may be expressed in the post-meiotic cells that are underrepresented in the larvae testes.

After estimating the average expression level (Figure 1b) and the percentage of expressed cells (Appendix A) within each type of cell for each gene, we annotate each gene as one of the five cell categories. According to unsupervised clustering, mitotic genes refer to those that are predominantly expressed in spermatogonia; meiosis genes refer to those that are predominantly expressed in spermatocytes, and post-meiosis genes refer to those that are predominantly expressed in spermatids. There are also some genes that span two adjacent stages or developing cell types (Figure 1b, e.g., ‘meiosis and mitosis’ genes that are expressed in both spermatogonia and spermatocytes). Overall, mitotic genes comprise the largest category (over 50% of the 15,189 genes expressed in the germline) among all five types of genes, with 6431 (42.3%) genes expressed in spermatogonia, and 1300 (8.6%) non-overlapping genes expressed in both spermatogonia and early spermatocytes, whereas genes predominantly expressed in spermatids comprise the smallest part, with only 690 genes (4.5%). Our division of genes according to their expressed cell types or stages is consistent with their respective enrichment of gene ontology (GO) terms (Appendix A). For example, mitotic genes are enriched for GOs of ‘mitotic cell cycle process’, ‘mitotic nuclear division’, etc., and post-meiotic genes are enriched for GOs of ‘spermatid development’ and ‘sperm motility’, etc.

Comparing with the autosomes excluding the dot chromosome, we found a significant (*p* < 0.05, Fisher’s test) enrichment of mitotic genes, but a significant (*p* < 0.05, Fisher’s test) underrepresentation of meiotic/post-meiotic genes, on both X chromosome and dot chromosome of *D. melanogaster*. This pattern is consistent with the results previously reported on the mouse X chromosome [30]. For the Y chromosome, over 50% of the genes are not expressed in any cell types, and it in contrast shows an underrepresentation of mitotic genes (Figure 1c,d) relative to the autosomes. A similar enrichment or under-representation pattern of certain testis genes on the X chromosome can be also observed, if we divide the genes according to the percentage of cells that they are expressed, among each cell type (Appendix A), suggesting our results are robust to the different methods of categorizing the genes. These patterns indicate a distinctive chromosomal composition of genes expressed in different testis cell types. They are consistent with the previous analyses using the larvae scRNA-seq data [24], which suggested that the dramatic reduction in X-linked gene expression starting from early until late primary spermatocytes, and a deficiency of X-linked meiotic genes is more likely to be caused by MSCI rather than lack of dosage compensation in meiotic cells. They are also consistent with the probably shared expression regulatory mechanism of X- and dot-linked genes during spermatogenesis [24].

### 2.2. New Genes of Different Ages or Produced by Different Mechanisms Have a Different Expression Pattern across Testis Cells

A large portion of functional new genes that become fixed in the population are usually expected to have evolved a diverged expression pattern from their parental genes to avoid functional redundancy [1]. After annotating the major expressed cell type(s) of each gene, we set out to compare between new vs. parental genes for their major expressed testis cell types to test this hypothesis. Our previous study comparing different age groups of new vs. parental genes uncovered the fact that new genes have gradually acquired active histone modification by their ages [8]. Consistent with this, the percentage of new genes expressed within at least one type of testis cells has increased by new gene age (Figure 2a, the parts that are not black). Additionally, the percentage of parental–new gene pairs that are expressed within completely non-overlapping cell types of testes becomes higher in the older groups of new genes (32~39%) than the new genes of younger ages (24% or 25% of group 5 or 6 genes, Figure 2a). Overall, new genes are significantly (*p* < 0.05, Fisher’s test) more enriched for meiosis or post-meiosis genes in the testis, but deficient for mitotic genes, compared with their parental genes (Figure 2a). The pattern of different expressed testis cell types is mainly attributed to those new genes that are of relatively older age (age group 1–3). These single-cell level expression patterns support the notion that new genes gradually acquired novel functions and diverge their expression patterns from their parental copies within testes during evolution. 

Particularly, there is a significant enrichment (*p* < 0.05, Fisher’s test) of mitotic genes among the parental genes of very young genes (age group 5 and 6 combined), which are specific to *D. melanogaster* or shared by *D. melanogaster* and *D. simulans*, relative to the genomic background (Figure 2a,c). We previously showed that these two groups of young genes are also more enriched for testis-specific genes relative to other age groups, whereas the ‘older’ new genes are more enriched for house-keeping genes [8]. These results together indicate that mitotic genes, or genes expressed in spermatogonia, have a disproportionately large role in producing the ‘out of the testis’ pattern of *Drosophila* new genes.

When we divided the new genes according to their origination mechanisms, namely DNA-based duplication, RNA-based duplication (retrotransposition) and de novo evolution from non-coding genomic DNAs, and also divided them by whether the parental and new gene copies reside on the same chromosome (intra-chromosomal) or not (inter-chromosomal); different categories of parental–new gene pairs show different expression patterns across testis cells. The differences of expressed testis cell types between parental and new genes are much larger for the new genes produced by retrotransposition compared with those produced by DNA-based duplication (Figure 2b), whereas there does not seem to be a pronounced difference between new genes produced by inter- or intra-chromosomal DNA duplications. Additionally, de novo genes also show a significantly different expression pattern across the testis cell types compared with the genomic background (Figure 2c). These patterns can be explained by the fact that retrogenes and de novo genes are much more likely to acquire novel *cis*-regulatory elements than DNA-based new gene duplications [1,2]. In particular, both new retrogenes and de novo genes exhibit an enrichment of meiosis or post-meiosis genes, i.e., genes expressed in spermatocytes and spermatids (‘into the meiosis’), and correspondingly an underrepresentation of mitosis genes (expressed in spermatogonia) compared with their progenitors or the genomic background (Figure 2b,c). Genes expressed in late spermatogenesis stages may be more frequently exposed to sperm competition than those expressed in spermatogonia, which probably accounts for the fixation of these new genes in the population.

### 2.3. MSCI Plays an Important Role in ‘out of the X’ Pattern of New Genes

Having discovered an ‘into the meiosis’ pattern for retrotransposed new genes and de novo genes, we further examine the parental–new genes pairs that are divided according to their chromosomal distributions. We hypothesize that if MSCI drives an ‘out of the X’ pattern of new genes, we expect to find an enrichment of meiosis genes for the new genes on the autosomes, whereas an underrepresentation of meiosis genes for their progenitors located on the X chromosome. This is indeed the case (Figure 3a, Appendix A, Appendix A): among 68 pairs of X-linked parental genes that produce an autosome new gene (‘X to A’, 41 pairs by DNA-based duplication, 27 pairs by retroposition), there are 42 autosomal new genes expressed in cells of meiosis or post-meiosis stages, in contrast to only 16 X-linked parental genes that are expressed at the same stages. Compared with the parental genes, the ‘X to A’ new genes are significantly (*p* < 0.05, Fisher’s test) enriched for meiosis and postmeiosis genes, but significantly deficient for mitotic genes. Such a significantly complementary expression pattern between parental and new genes is not found for other inter-chromosomal parental–new gene pairs between autosomes (‘A to A’) or from autosomes to X chromosome (‘A to X’) and provides direct evidence for MSCI in *Drosophila* (Figure 3a). 

To further delineate the direction of, and measure the extent of germline cell type changes of each new gene compared with its parental gene, we labelled the four germline cell categories or their combinations (‘mitosis’, ‘mitosis and meiosis’, ‘meiosis’, ‘meiosis and postmeiosis’ in Figure 1b) as 1 to 4. For example, if a parental gene predominantly expressed in spermatogonial or mitotic cells (cell type 1) produces a new gene predominantly expressed in spermatids or meiotic/postmeiotic cells (cell type 4), such a gene pair is annotated with an ‘into the meiosis’ change of 3. Any positive values of cell type change for a parental gene expressed in the mitotic cells (cell type 1 or 2) are defined with an ‘into the meiosis’ change, vice versa for the ‘into the mitosis’ change with negative values, or others with no or minor changes as ‘within the meiotic cells’ or ‘within the mitotic cells’. In general, new genes are always much more likely to show a propensity of ‘into the meiosis’ change than the opposite direction of ‘into the mitosis’ change when being compared with their parental genes, except for DNA-based new gene duplications between autosomes. Additionally, retrotransposed new genes are more likely to show an ‘into the meiosis’ change relative to DNA-based new gene duplication of the same inter-chromosomal gene duplication category. For example, there is about 1.5-fold (59.2% vs. 40.5%) more ‘X to A’ retrogenes that exhibit a ‘into the meiosis’ change than the ‘X to A’ DNA-based new gene duplicates. In fact, ‘X to A’ retrotransposed new genes exhibit the highest percentage of ‘into the meiosis’ change among all types of inter-chromosomal gene duplications (Figure 3b). These data together support the notion that MSCI drives the ‘out of the X’ pattern of new genes in *Drosophila* [7,11].

### 2.4. New Gene Cases of ‘into the Meiosis’

We finally examine the parental–new gene pairs that have been previously studied for their functions, in order to elucidate the evolutionary forces driving their expression divergence across different testis cells (Appendix A). Among the ‘X to A’ parental–new gene pairs that exhibit the ‘into the meiosis’ trend, we find nuclear transport factor 2 (*Ntf-2*) and its retroposed new gene *Ntf-2r* that was identified nearly 20 years ago as one of the first reported retroposed new genes [31]. The coding sequence of *Ntf-2r* has been characterized to have undergone positive selection relative to its parental copy, and its promoter was inferred to drive a late testis-specific expression similar to that of *beta-2 tubulin*. This is indeed the case: while *Ntf-2* concentrates its expression within the spermatogonia, the expression of *Ntf-2r* has the highest level within early spermatocytes and extends further into spermatids (Figure 4). A second case is *sex lethal* (*Sxl*) and its duplicated gene copy *CG5213* or *maca*. *Maca* has been recently reported to be critical for the proper splicing of Y-linked genes *kl-2*, *kl-3* and *kl-5* with gigantic introns, and the *maca* mutant exhibits a failure in the sperm individualization process and thus male sterility [32]. This is also consistent with *maca*’s expression concentrated within the early spermatocytes, compared with *Sxl* that is predominantly expressed in spermatogonia. Among the ‘A to A’ genes that exhibit a ‘into the meiosis’ trend, we found two cases of proteasome new genes *Prosalpha4T1* and *Prosalpha3T* that have been proposed to be important for sperm tail elongation. The parental genes of both cases are housekeeping genes that are expressed in most tissues, and in testes, predominantly in spermatogonia, whereas the two new genes are exclusively expressed during late spermatogenesis, and in few or no other adult tissues. Mutants of *Prosalpha4T1* have been recently shown to cause male sterility [33]. These cases indicate that many new genes have acquired novel functions through evolving divergent cell-level expression from their parental genes in testes.

We do notice that there are also cases of ‘into the meiosis’ among the ‘A to X’ genes (Figure 3b); this suggests that the MSCI does not impact all the genes located on the X chromosome, particularly those whose expression is required for spermatogenesis. Although underrepresented, there are still many X-linked genes expressed during meiosis (Figure 1b). Among all the identified cases of ‘A to X’ genes with the ‘into the meiosis’ change (Appendix A), we have not found cases whose functions have been studied in detail. However, ‘A to X’ duplicated new genes, or tandem gene duplications on the X chromosomes that show predominant expression within meiotic cells for both parental and new gene copies can provide us some insights. Among these cases, we found the X-linked *xmas-1*, whose parental gene *Mmp2* is located on the autosomes. *Mmp2* is an important developmental gene related to metamorphosis and is biasedly expressed in pupae but has a low expression level in testes [34]. Interestingly, it is also required for ovulation and specifically expressed in follicle cells [35,36], whereas *xmas-1* encodes a hydrophilic protein required for spermatogenesis and oogenesis and one of its mutant lines shows male sterility [37]. This probably represents another case of gene duplication that evolved to resolve sexual conflict [38], in addition to the recently reported case of *Apollo* and *Artemis* gene duplication pair [39], whereas the tandemly duplicated genes of *Sdic* family on the X chromosome are mainly expressed in meiotic cells (Figure 4), and have been suggested to be important for sperm competition by influencing the sperm motility [40]. These cases together suggest that, besides the selection of MSCI against meiotic genes to be located on the X chromosome, sperm competition and sexual conflict may nevertheless fix some X-linked new genes that are expressed during meiosis. More future functional studies into these ‘A to X’ new genes that show meiotic gene expression (Appendix A, Appendix A) can test this hypothesis.

## 3. Discussion

Here, we harness the recently published scRNA-seq data of *D. melanogaster* and address several important questions on the origin and evolution of new genes in testes, in the context of distinct regulatory programs of X chromosome vs. autosomes. In somatic cells of male *Drosophila*, male-specific lethal (MSL) protein complex specifically targets the X chromosome and results in a balanced expression level between the single-copied X chromosome vs. the autosomes with two copies [41]. Such a canonical dosage compensation mechanism is absent in the male germline cells, and seems to be replaced by a non-canonical dosage compensation whose mechanism is still unclear, in spermatogonia [42]. Previous transcriptomic and epigenetic studies of enriched spermatogonia cells of *bam* mutant flies [43,44] showed an overall enrichment of chromatin remodelling factors, and a distinct histone modification pattern from mammalian embryonic stem cells on genes related to cell differentiation. These factors may explain genes predominantly expressed in spermatogonia are most abundant among all genes compared with those expressed in other testis cells (Figure 1), suggesting a more permissive transcriptional environment in spermatogonia than other testis cells. It may also explain that mitotic genes act as an important source for generating the young genes and contributing to the ‘out of the testis’ pattern of new gene origination in *Drosophila*. Particularly, mitotic genes are enriched on the *Drosophila* X chromosome (Figure 1), and this has been also reported on the mouse X chromosome [30,45]. The convergent accumulation of male germline mitotic genes on the two independently evolved X chromosomes probably reflects the ‘fast-X evolution’ effect, that male recessive beneficial alleles are expected to be fixed at a faster rate on the X chromosome than the autosomes [46,47]. 

On the other hand, the general deficiency of male meiotic genes on the X, and consistently, an excess of X-borne new genes onto the autosomes (‘X to A’ genes) that are predominantly expressed in the male meiotic cells (Figure 3), demonstrate the effect of MSCI in *Drosophila*. Again, a similar underrepresentation of male meiotic genes on the mouse X chromosome has also been reported before [30]. Such similar patterns of the X chromosomes of mouse and *Drosophila* indicate the selection of MSCI against X-linked genes that are specifically expressed during meiosis. This does not preclude some X-linked new genes that evolved novel or necessary functions during meiosis to be fixed in population (e.g., *Sdic* gene family), driven by either sperm competition or sexual conflict, which is expected to be more intense during late spermatogenesis. In mammals, the contrasting forces of ‘fast-X evolution’ and MSCI together resulted in an excess of genes duplicated onto the X chromosome, as well as an excess of genes relocated from the X chromosome (the ‘gene traffic’ pattern) [12]. The development of scRNA-seq data atlas of different organs of both *Drosophila* and mammals [48], as well as future functional studies into the parental–new gene pairs that have been annotated for their divergent expressed testis cell types in this study (Appendix A), provide more insights into the complex molecular and evolutionary factors acting on the X chromosome, and the evolutionary trajectories of new genes.

## 4. Materials and Methods

### 4.1. Datasets of New Genes and scRNA-seq

The new gene dataset of *Drosophila* is derived from GenTree [27]. We only used those identified de novo genes or new genes with clearly defined parental genes with respective age information. Among these, there are 565 new genes produced by gene duplication (464 from DNA-based duplication and 101 from RNA-based duplication) and 140 de novo genes. These new genes are divided into 6 age groups based on their origination time, with the youngest group only restricted to *Drosophila melanogaster* and labeled as branch 6 and so on (Figure 1a). For duplication pairs, the new genes were shorter than their parental genes in gene length, and have less exons than their parental genes, especially for RNA-level duplicated new genes, as retrogenes were generated from mRNA and had no introns (Appendix A). Additionally, for the chromosomal location of new genes, it seems not having specificity in a chromosome or distance to TE (Appendix A). It was reported that new genes in *Drosophila* have an ‘out of X’ pattern. For our new gene set, the inter-chromosomal RNA-level duplication new genes indeed have an ‘out of X’ pattern, showing an excess of new genes retroposed from X to autosomes (X-squared = 5.7022, *p*-value = 0.01694) when being compared with the expected ratio (calculated basing on gene number of source chromosome, chromosome size of targeted chromosome and frequency of retroposition to a given chromosome [7]) (Appendix A).

We used previously published scRNA-seq data of adult *D. melanogaster* testes from 2 different strains [26]. Reads were aligned and filtered based on the FlyBase dmel_r6.15 reference genome with Cellranger 4.0.0 [49], and count matrices were generated. Potential doublets were detected using DoubletFinder [50] and removed. R package Seurat 4.0.2 [51] was used for single-cell downstream analysis. First, genes expressed in at least 3 cells and cells with at least 200 genes expressed were kept. Remaining cells were normalized by SCTransform [52] with default parameters. A total of 2000 variable features were selected in each dataset, and 2 datasets were integrated using the canonical correlation analysis (CCA) method implemented in Seurat by default augments. After integration, FindNeighbors and FindClusters were performed with 30 PCs and the resolution was 2. Cells were visualized with Uniform Manifold Approximation and Projection (UMAP). We annotated 7 cell types, including 4 germline cell types and 3 somatic cell types with markers used in [26]. We used *AGO3* and *Aub* as markers for spermatogonia. Clusters with high expression of *CycB* and *fzo* were annotated as early spermatocytes and clusters expressing *twe* but not *fzo* were identified as late spermatocytes. Spermatids were identified by *soti*, *boly* and *p-cup*. Clusters enriched in *tj* and *eya* are cyst cells. Hub cells are marked by *Fas3* and *zfh1*. Cells expressing *MtnA* but not *zfh1* are epithelial cells. Average expression levels (log2(CPM + 1)) for each cell type were calculated. Since germline somatic cells are of relatively low numbers, we focused our study on germ cells.

### 4.2. Defining the Major Expressed Cell Type of Drosophila Genes

For all the genes annotated in Flybase (r6.15), 15,189 genes were expressed in germline cells, 30 genes were only detected in somatic cell types, and the other 2508 genes were not expressed in testis. We used the 15,189 germline expressing genes (log2(CPM + 1)) to generate a row-scaled heatmap with the ComplexHeatmap (version 2.6.2 [53]) package in R, using hierarchical clustering (hclust in r, the complete linkage method) and cutting the generated dendrogram tree into 7 clusters (cutree function called by color_branches from dendextend package in r) basing on the gene expression pattern (the blank lines in Figure 1b are the borders of the clusters). Based on the development of the germline cells and clustering patterns, we divided 7 clusters into 5 categories, genes enriched in spermatogonia were defined as mitosis genes, genes enriched in spermatocytes were defined as meiosis genes, genes enriched in spermatids were defined as post-meiosis genes. We further defined those enriched in both spermatogonia and early spermatocytes as mitosis and meiosis genes, and those enriched in spermatocytes and spermatids as meiosis and post-meiosis genes. Since the post-meiosis stage contains very few genes in the clustering result, we combined the meiosis and post-meiosis stage together. To verify our categorization of genes, we not only clustered the expressional level using row-scaled log2(CPM + 1) value of each gene in every cell type, but also used the percentage of expressed cells in each cell type to carry out hierarchical clustering. The overall pattern of different types of genes across different chromosomes stayed the same, regardless how we clustered the genes (Appendix A).

### 4.3. Comparing the Percentage of Genes in Different Stages between Chromosomes

To find out if there was enrichment of genes in different stages in each chromosome, first we showed the distribution of different categories of genes in autosomes, 4, X and Y chromosomes, and compared whether the difference of percentage for genes in each stage between other chromosomes (4,X,Y) and autosomes were significant (Figure 1c). The gene numbers in each chromosome in each stage were shown in Appendix A.

For the calculation method of the ratio in Figure 1d, we first normalized the gene numbers in each stage with the gene numbers in each chromosome and obained the normalized numbers in different stages in each chromosome (Figure 1c, Appendix A), then calculated the ratio between other chromosomes vs. autosomes based on the normalized numbers in each stage of each chromosome (Figure 1d, Appendix A).

### 4.4. Comparing the Expressed Cell Types between the New and Parental Genes

To measure the stage shift between the new and parental genes, we proposed a calculation method. From spermatogonia to spermatids, corresponding to numbers 1 to 4, the difference between child value and parent value is the pairs’ stage shift condition. The negative number means the child stage is before the parent stage, and a positive number means the child stage is after the parent stage, thus we obtained a value by which to measure the stage shift condition. All the significant tests in main figures were performed using the Fisher test in R (Appendix A). For duplicated new genes, the tests were checked between parent and child. For de novo new genes, the tests were checked between de novo new genes and all annotated genes. We used Flybase (http://flybase.org/, accessed on 4 May 2021) to manually check individual genes’ expression patterns across different tissues or stages, based on the modENCODE transcriptome data [54,55].

## 5. Conclusions

In conclusion, we provide cell-level evidence for MSCI driving the new genes off the X chromosome to be relocated onto the autosome in Drosophila, and uncovered the different expression patterns of new genes across testis cells compared to their progenitors.

## Figures and Tables

**Figure 1 genes-12-01796-f001:**
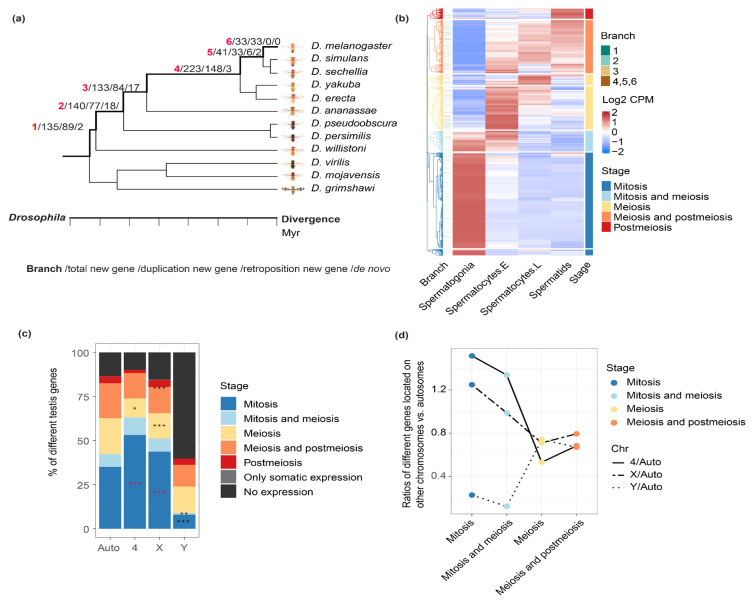
Comparing chromosomal distribution of genes across stages of spermatogenesis. (**a**) The number of new genes divided by branches and origination mechanisms, the description of each number separated by slash is shown in the bottom, the divergence time of each branch is also shown in the plot with number 1 to 6. New genes of branch 6 are the youngest new genes and only restricted to *Drosophila melanogaster.* New genes of branch 5 are shared by *D. melanogaster*, *D. simulans* and *D. sechellia* and ‘older’ than those of branch 6, and so on. (**b**) Unsupervised clustering of testis-expressing genes according to their average expression level within each of the four types of cells (spermatogonia, early spermatocytes, late spermatocytes and spermatids). Each line is a gene, and each column shows the color scaled normalized expression level, i.e., the log2(CPM + 1) value per gene within a testis cell type. The genes are then divided into 5 categories based on their predominant expressed cell types as mitotic genes, meiotic genes, postmeiotic genes or genes that span two adjacent stages. We also show the branch information of new genes next to the heatmap, with most genes shown in white color as they are old genes shared by all *Drosophila* species. (**c**) The distribution of different categories of genes in autosomes, 4, X and Y chromosomes. The black asterisks mean significant underrepresentation when being compared with autosomes, whereas the red asterisks means significant enrichment. Since there are few post-meiosis genes, we combine them with meiosis genes when performing the statistical tests, and the asterisks are shown between the orange and red. (**d**) Each dot shows the ratio of numbers of genes on the X/Y/dot chromosome vs. autosomes within each cell category. “*”: *p* < 0.05, “**”: *p* < 0.01, “***” *p* < 0.001.

**Figure 2 genes-12-01796-f002:**
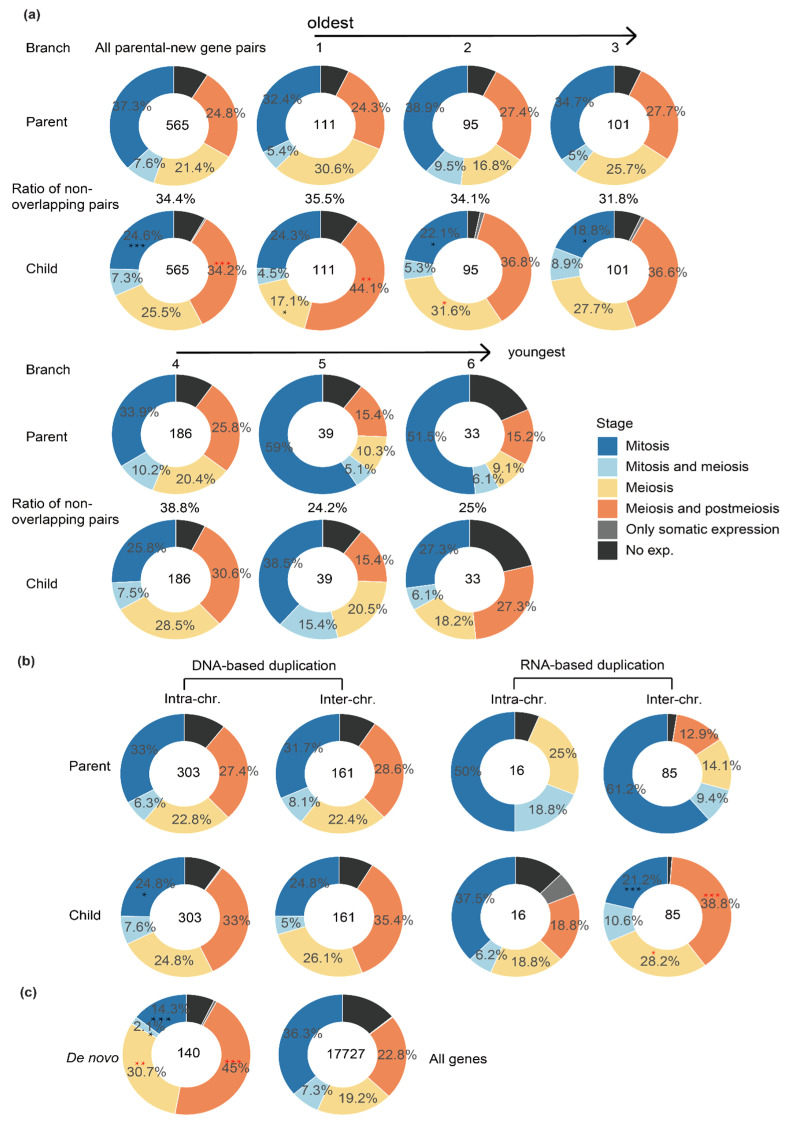
The expression patterns of parental–new gene pairs across testis cells divided by their age and origination mechanisms. (**a**) Comparing parental (‘Parent’) and new (‘Child’) genes of different age groups (1–6, Figure 1a) for their expression patterns within different cell types. Different colors represent different categories of genes inferred from Figure 1b. The numbers within each ring shows the number of parental–new gene pairs of each age group. Additionally, the percentage number shown between the rings of parental genes and new genes indicates the percentage of genes that are expressed in completely non-overlapping cell types of testes. (**b**) Comparison between DNA-based (‘DNA’) and RNA-based (‘RNA’) gene duplications and also between intra- or inter-chromosomal duplications. (**c**) Comparison between the de novo genes and the genomic background of all genes (Flybase 6.15 released version). The significant tests were performed by Fisher test for (**a**,**b**), between each parental and new gene set. Red asterisks mean significantly enriched; black asterisks mean significantly underrepresented. “*”: *p* < 0.05, “**”: *p* < 0.01, “***” *p* < 0.001.

**Figure 3 genes-12-01796-f003:**
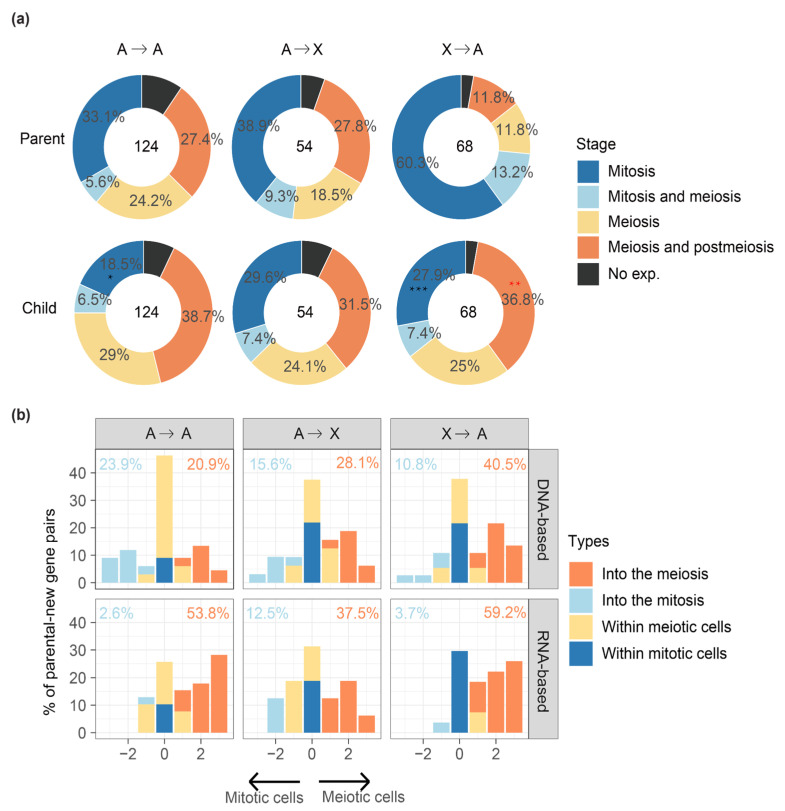
Comparing the testis cell type differences between parental and new genes. (**a**) Comparing the testis cell composition between the parental and new genes divided by their relative chromosomal positions. We divided the parental–new gene groups into: ‘A to A’ (an autosomal parental gene produces an autosomal new gene), ‘A to X’ (an autosomal parental gene produces an X-linked new gene), ‘X to A’ (an X-linked parental gene produces an autosomal new gene). Red asterisks mean significantly enriched; black asterisks mean significantly underrepresented. “*”: *p* < 0.05, “**”: *p* < 0.01, “***” *p* < 0.001. (**b**) We further cross-examine the three types of inter-chromosomal parental–new gene pairs regarding their different origination mechanisms as either DNA-based or RNA-based duplications. The x-axis shows the extent and direction of testis cell type changes between the parental and new genes. We labelled the genes annotated to be predominantly expressed at one or two types of germline cells (Figure 1b) from spermatogonia to spermatids as 1 to 4. If a parental gene that is expressed mainly in spermatogonia (cell type 1) or both spermatogonia and early spermatocytes (cell type 2) shows a positive value of cell type change for its new gene, it is defined as showing an ‘into the meiosis’ change. Similarly, we can also define cell type changes as ‘within the meiotic cells’ or ‘within the mitotic cells’ if the new and parental genes differ by 1 or less, and as ‘into the mitosis’ if the difference is a negative value and the new gene is predominantly expressed in spermatogonia (cell type 1 or 2). We show the percentage of ‘into the meiosis’ gene pairs in orange, and that of ‘into the mitosis’ in blue for each category.

**Figure 4 genes-12-01796-f004:**
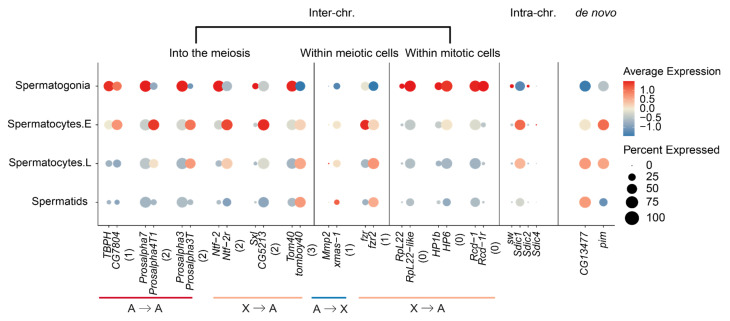
New gene cases of ‘into the meiosis’. We show each pair of parental (**left**) and new (**right**) genes next to each other for their color-scaled expression levels and percentage of expressed cells scaled by the bubble size within each type of testis cell. Genes are selected based on their previous functional studies. The number below each gene pair was the extent and direction of testis cell type changes between the parental and new genes, which was defined in Figure 3b.

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
