# Peer review of "On the Origin and Evolution of Drosophila New Genes during Spermatogenesis"

_genes, 2021, doi:10.3390/genes12111796_

Round 1

Reviewer 1 Report

This study uses single-cell RNA-seq data to examine whether meiotic sex chromosome inactivation drives the evolution of new duplicate genes in Drosophila. The study is well-designed and clever, and the results are interesting. I only have a couple of comments that I would like the authors to address:

1. For Figure 2, it would be helpful if there was an arrow at the top indicating that age 1 is oldest and age 6 is youngest. 

2. For Figure 3, are numbers of A->A, X->A, and A->X consistent with what one might expect based on numbers of genes on autosomes and sex chromosomes? 

3. For k-means clustering, I assume that k=5? This is not clear in the methods. Also, how many times was each k-means clustering algorithm run?

4. More details are needed for the Fisher's exact tests performed. 

Author Response

The first reviewer

  1. All the figures are small in size, so they are difficult to see. Please enlarge the figures, and it would also be better to make the letters and numbers larger.

A: We apologize for the issue, and have made all figures larger in the revised ms.

  1. Could the authors give some information (summary) about the genes that they analyzed? For instance, what are the size (kbp) of the genes. Do these genes have introns? Do you find any tendency on the newly born genes?

A: The Drosophila new genes are significantly (P<0.05, Wilcoxon rank sum test) s  than their parental genes in gene length, and have less exons than their parental genes, because of retrotransposition or partial gene duplication (Supplementary Fig. S1). We added this summary onto line 93 of the revised text and showed the added Supplementary Fig. 1 below.

  1. Related to above question. Is there any information about the chromosome location of the newly born genes? Are they close to transposable elements, meiotic recombination hotspots, etc. or are these genes just inserted randomly within the genome?

A: We added another Supplementary Fig. 2 at line 101, and below to address the reviewer’s question. Our results show that the young new genes(branch 5,6) have a significant higher ratio located on the X chromosome when being compared to autosomes (Fisher’s exact test, p = 2.492e-06 for branch 5 and 7.498e-06 for branch 6), while old new genes(branch 1) have a significant higher ratio in autosomes when compared to X chromosome (Fisher's exact test, p = 0.01972). This is consistent with the previous result (Zhang et al. 2010 Genome Res.). For the distance between TE(flybase r6.15) and new genes, it did not show a significant difference from those between TE and all the other genes in the genome. However, it has been previously reported that enrichment of the transposable element DNAREP1 family nearby the new genes that are distantly located on the same chromosome, or on the different chromosomes compared to their parental genes (Shuang et al. 2008 PloS Genetics). So it is likely that certain repeats have mediated the duplication and relocation of the new gene. For the distance between meiotic recombination hotspots(Chan et al. 2012) and new genes, it also didn’t show significant difference between de novo originated new genes, RNA-based new genes, DNA-duplication new genes and the background genes. We added these information into the revised text when necessary. However, since the major aim of this project is not to deal with the general property of the new genes, we did not extensively talk about these points.

Reviewer 2 Report

In this report, Su et al. propose how new genes evolve during spermatogenesis, by studying scRNA-seq data from different types of Drosophila species. Their findings are interesting .  I have some minor comments.

  1. All the figures are small in size, so they are difficult to see. Please enlarge the figures, and it would also be better to make the letters and numbers larger.
  2. Could the authors give some information (summary) about the genes that they analyzed? For instance, what are the size (kbp) of the genes. Do these genes have introns? Do you find any tendency on the newly born genes?
  3. Related to above question. Is there any information about the chromosome location of the newly born genes? Are they close to transposable elements, meiotic recombination hotspots, etc. or are these genes just inserted randomly within the genome?

Author Response

The second reviewer

  1. For Figure 2, it would be helpful if there was an arrow at the top indicating that age 1 is oldest and age 6 is youngest. 

A: I have added the arrows in fig2.

  1. For Figure 3, are numbers of A->A, X->A, and A->X consistent with what one might expect based on numbers of genes on autosomes and sex chromosomes?

A: For RNA-level duplication, the numbers of A->A, X->A, and A->X was significantly different from the expected (X-squared = 6.4912, p-value = 0.03894), and there is a significant excess of new genes retroposed from X to autosomes (X-squared = 5.7022, p-value = 0.01694). For DNA-level duplication, the numbers of A->A, X->A, and A->X was not significantly different from the expected (X-squared = 3.1788, p-value = 0.204), and there was a higher AtoX ratio than the expected but not significant (X-squared = 3.0951, p-value = 0.07853).

I tested if the observed numbers of A->A, A->X, and X->A were consistent with the expected proportion using the calculation method by Maria (Vibranovski et al. 2009) to calculate the expected number of inter-chromosomal gene movements, which considering the influences of gene number of source chromosome, chromosome size of targeted chromosome and frequency of retroposition(RNA)/duplication(DNA) to a given chromosome. The gene number of source chromosome and chromosome size of targeted chromosome were obtained from annotation of D. melanogaster in flybase(r6.15); Frequency of retroposition(RNA)/duplication(DNA) to a given chromosome was from Maria D. Vibranovski (for RNA-level duplication, 0.75 for X and 1 for autosomes; for DNA-level duplication, 1/2 for both X and autosomes, Supplementary Table S4).

  1. For k-means clustering, I assume that k=5? This is not clear in the methods. Also, how many times was each k-means clustering algorithm run?

We used hierarchical clustering rather than k-means clustering with the complete linkage method to generate the heatmap (hclust in r), and then used cutree (called by color_branches from dendextend package in r) to cut the tree into 7 clusters. We didn't have a preset cluster number but decided the cluster number based on expression pattern and then classified the clusters to different groups based on development of cell types. That is the reason that we used hierarchical clustering rather than k-means clustering as k-means clustering need a preset k value. Here I showed the original heatmap with the cluster number added to each cluster (the blank lines are the borders of clusters). Finally, I combined cluster1 and cluster2 to the mitosis group as the two clusters are mainly enriched in spermatogonia, and combined the cluster4 and cluster5 to meiosis group as the two were enriched in spermatocytes.

  1. More details are needed for the Fisher's exact tests performed. 

I have added Fisher's exact tests in Supplementary Table S3.

Reviewer 3 Report

The analyses in Su et al. are complex and thorough. However, their rationale is vague, and the logical flow from their results to their conclusions is hard to follow. Their conclusions on the implications of the gene expression patterns they observe are not well sustained by the data they present. 

Author Response

The third reviewer

  1. The analyses in Su et al. are complex and thorough. However, their rationale is vague, and the logical flow from their results to their conclusions is hard to follow. Their conclusions on the implications of the gene expression patterns they observe are not well sustained by the data they present. 

We apologize for the issues that are raised by the reviewer, and we have improved our manuscript during this revision according to other reviewers’ questions. We wonder if the reviewer could specify the parts in the ms that he/she felt problematic, and we can further improve or clarify.

Round 2

Reviewer 1 Report

The authors have satisfactorily addressed most of my concerns. I do not have any further comments about the manuscript

Author Response

Thank you for the comments.

Reviewer 3 Report

After revisiting the comments from other two reviewers, and a careful re-read of the manuscript along with some literature, I came to the conclusion that I did not correctly understand the authors’ approach in my first contact with the manuscript. Nevertheless, I still believe some points need clarification, and sometimes the logic behind the claims of this article is weak.

Please revisit these points:

  • Page 2, first paragraph: Try to explain the relocation of testis-biased genes out of the X more clearly. This paragraph goes back and forth to different ideas, when it can be much more simplified. Also, selection against sexually antagonistic interactions is not “a second factor”, it would be the force that mediates the inactivation of the X in males and the removal or relocation of male-biased genes from the X. It is unclear why “the lack of canonical dosage compensation of X chromosome in the male germline” is relevant here. The paper the authors cite about dosage compensation (i.e. # [18]), also reports no evidence of X chromosome-specific inactivation during meiosis. Please discuss.
  • Specifically, if MSCI exists in Drosophila and drives the relocation of X-borne new genes onto the autosomes, we would expect that a shift of autosomal new genes to be expressed toward the middle or late spermatogenesis (e.g., in spermatocytes or spermatids) stages, relative to their parental genes that are expressed in the early spermatogenesis (mitotic) stage (e.g., spermatogonia)” Please, explain why would we expect that, and why would that evidence the existence of MSCI.
  • Please, specify the total number of genes considered for Fig1 C and D. Chromosomes 4 and Y are small and poor in genes, and therefore few genes would make up to a big percentage. Also, please clarify what is on the Y axis in Figure 1 D, the values on the Y axis are confusing.
  • They are also consistent with the probably shared expression regulatory mechanism of X- and dot-linked genes during spermatogenesis.” Include a cite.
  • These patterns can be explained by the fact that retrogenes and de novo genes are much more likely to acquire novel cis-regulatory elements than DNA-based new gene duplications.” Include a cite.
  • In fact, ‘X to A’ retrotransposed new genes exhibit the highest percentage of ‘into the meiosis’ change among all types of interchromosomal gene duplications (Figure 3b). These data together support the notion that MSCI drives the ‘out of the X’ pattern of new genes in Drosophila” It is true thatX to A’ retrotransposed new genes exhibit the highest percentage of ‘into the meiosis’ change” (59.2%), but it is not that different from the percentage of “into the meiosis” change for A to A retrotransposed genes (53.8%). From Figure 3b, what I see is a general “into the meiosis” trend for new genes originated by RNA-based mechanisms, for all types of interchromosomal gene duplications, and also for DNA-based gene duplications X to A. I am not sure these data support that MSCI drives the “out of the X” pattern.
  • The existence of A to X new genes that shows “into the meiosis” pattern contradicts the author´s model, or the existence of a general MSCI of the X during meiosis. They acknowledge this caveat: “We do notice that there are also cases of ‘into the meiosis’ among the ‘A to X’ genes (Figure 3b), this suggest that the MSCI does not impact all the genes located on the X chromosome, particularly those whose expression is required for the spermatogenesis.” If these genes can stay in the X and avoid silencing, then the selective pressure that pushes testis-biased expressed genes out of the X disappears. And thus, these data do not support that MSCI drives the “out of the X” pattern.  

Author Response

The third reviewer

After revisiting the comments from other two reviewers, and a careful re-read of the manuscript along with some literature, I came to the conclusion that I did not correctly understand the authors’ approach in my first contact with the manuscript. Nevertheless, I still believe some points need clarification, and sometimes the logic behind the claims of this article is weak.

We would like to emphasize that MSCI in Drosophila is a question that has been under debate in the last 10 years and only until recently there is new mounting evidence supporting its existence in Drosophila through the single-cell RNA-seq technology and novel chromosome territorial study. The reviewer seems to have a tendency to accept one simple solution for a question as complicated as MSCI, which unfortunately is not the case, similar to many other intriguing questions in science. For example, the reviewer thinks ‘sexually antagonistic selection is the force that mediates the inactivation of the X’. And also, there seem to be some results ‘contradict’ the existence of MSCI. I would respectfully disagree that there are logic issues behind our results; they merely reflect the complex forces acting on the X-linked male-biased genes, which we have already elaborated in both introduction and discussion in the ms. And I address the reviewer's question point-by-point as below.

  1. Page 2, first paragraph: Try to explain the relocation of testis-biased genes out of the X more clearly. This paragraph goes back and forth to different ideas, when it can be much more simplified. Also, selection against sexually antagonistic interactions is not “a second factor”, it would be the force that mediates the inactivation of the X in males and the removal or relocation of male-biased genes from the X. It is unclear why “the lack of canonical dosage compensation of X chromosome in the male germline” is relevant here. The paper the authors cite about dosage compensation (i.e. # [18]), also reports no evidence of X chromosome-specific inactivation during meiosis. Please discuss.

A: In the paragraph that was mentioned by the reviewer, we first described MSCI, then sexually antagonistic selection as ‘the second factor’. Such a description does not suggest sexually antagonistic selection is less or more important, simply indicates the fact that there are multiple factors that account for the relocation of X-linked male biased genes. We would also like to correct the reviewer here, that sexually antagonistic selection is ‘one of the forces’, rather than ‘the force’ that mediates the relocation of male-biased genes from the X. Although sexually antagonistic selection has been invoked to account for X-inactivation in males, it is a hypothesis (‘SAXI hypothesis’, Wu and Xu 2003 Trends in Genetics), yet it has been established that X inactivation or meiotic sex chromosome inactivation can also evolve to silence the transposable elements that are located at the unpairing regions between the X/Y, as a special form of meiotic silencing of synapsed chromatin (Turner 2007). Therefore, there are factors other than sexually antagonistic selection that can account for the evolution of MSCI. Hence we also think it is not accurate to say sexually antagonistic selection ‘is the force that mediates the inactivation of the X in males’.

Regarding the ref. 18, we already pointed out in the next paragraph: ‘In addition, debates about whether there is MSCI in Drosophila [22,23] have lasted until very recently.’ As mentioned by the reviewer, ref. 18 reported there is no evidence of MSCI during meiosis. However, this was refuted by the later studies [ref. 22, 23], then we cited a more recent study based on single-cell RNA-seq [ref. 24] that provided more mounting evidence of the existence of MSCI in Drosophila. This has all been discussed at line 64-74 on page 2.  We would like to point out that it is a norm that many scientific results are under debate and there is really no single answer, or can be simplified. Particularly regarding the relocation of X-linked male-biased genes in Drosophila that was studied here, over ten years ago, there was a paper titled ‘A complex suite of forces drives gene traffic from Drosophila X chromosomes’ (Meisel et al. 2009 GBE). The title speaks for itself.

  1. “Specifically, if MSCI exists in Drosophila and drives the relocation of X-borne new genes onto the autosomes, we would expect that a shift of autosomal new genes to be expressed toward the middle or late spermatogenesis (e.g., in spermatocytes or spermatids) stages, relative to their parental genes that are expressed in the early spermatogenesis (mitotic) stage (e.g., spermatogonia)” Please, explain why would we expect that, and why would that evidence the existence of MSCI.

A: MSCI obviously occurs during meiosis, i.e., in spermatocytes or spermatid cells but not in spermatogonia (mitotic) cells. Therefore if it indeed drives the relocation of male-biased genes in Drosophila, we expect the X-linked male-biased parental genes to be underrepresented in meiotic cells, while autosomal new genes to be overrepresented in meiotic cells. We revised the text on line 81 of page 2 to make it clearer.

  1. Please, specify the total number of genes considered for Fig1 C and D. Chromosomes 4 and Y are small and poor in genes, and therefore few genes would make up to a big percentage. Also, please clarify what is on the Y axis in Figure 1 D, the values on the Y axis are confusing.

A: We have now added the gene numbers in Fig1c to Supplementary Table S5a, and detailed ratio values in Fig1d to Supplementary Table S5c. Indeed, there are much fewer genes on the chr4 and chrY, but this was accounted for in figure 1c, as we showed here the ratio (i.e., genes of each category vs. the total numbers of genes on that chr) on the y-axis. And also in figure 1d, we also showed a ratio on Y-axis, which is divided by the ratio of figure 1c of other chromosomes. vs. autosomes. We described this part in more details in the method during this revision.

  1. “They are also consistent with the probably shared expression regulatory mechanism of X- and dot-linked genes during spermatogenesis.” Include a cite.

A: We added the citation (Mahadevaraju et al. 2021)

  1. “These patterns can be explained by the fact that retrogenes and de novo genes are much more likely to acquire novel cis-regulatory elements than DNA-based new gene duplications.” Include a cite.

A: We added the citation (Chen et al. 2013)(Long et al. 2003)

  1. “In fact, ‘X to A’ retrotransposed new genes exhibit the highest percentage of ‘into the meiosis’ change among all types of interchromosomal gene duplications (Figure 3b). These data together support the notion that MSCI drives the ‘out of the X’ pattern of new genes in Drosophila” It is true that “‘X to A’ retrotransposed new genes exhibit the highest percentage of ‘into the meiosis’ change” (59.2%), but it is not that different from the percentage of “into the meiosis” change for A to A retrotransposed genes (53.8%). From Figure 3b, what I see is a general “into the meiosis” trend for new genes originated by RNA-based mechanisms, for all types of interchromosomal gene duplications, and also for DNA-based gene duplications X to A. I am not sure these data support that MSCI drives the “out of the X” pattern.

A: It is true that ‘A to A’ retrogenes but not DNA-based new gene duplications also exhibit an excess of ‘into the meiosis’ pattern. This is consistent with figure 2B, as we described in line 215-234 of page 8, that it is a general property of retrogenes that are more likely than the DNA-based duplications, to show a different expression pattern than the parental genes. Here the different expression pattern in the context of testis cells means, the retrogenes tend to be expressed in meiotic cells, while the parental genes tend to be expressed in mitotic cells.

However, there is also a pronounced ‘into the meiosis’ pattern among the DNA-based ‘X to A’ but not ‘A to A’ parental-new gene pairs (Figure 3b, upper panel). This pattern has nothing to do with the general property of retrogenes, and supports that there are indeed critical differences between autosomes and X chromosomes, and MSCI is probably driving the relocation of X-linked male-biased genes. So the pattern that we observed for ‘A to A’ retrogenes can be explained by the fact that retrogenes are more likely to recruit novel cis-regulatory elements, and become fixed in meiotic cells driven by sperm competition. These points have already been mentioned in the previous version of ms.

  1. The existence of A to X new genes that shows “into the meiosis” pattern contradicts the author´s model, or the existence of a general MSCI of the X during meiosis. They acknowledge this caveat: “We do notice that there are also cases of ‘into the meiosis’ among the ‘A to X’ genes (Figure 3b), this suggest that the MSCI does not impact all the genes located on the X chromosome, particularly those whose expression is required for the spermatogenesis.” If these genes can stay in the X and avoid silencing, then the selective pressure that pushes testis-biased expressed genes out of the X disappears. And thus, these data do not support that MSCI drives the “out of the X” pattern.

A: As we have discussed, there is a complex suite of forces acting on the male-biased genes on the X chromosome, or the rest of the genome. MSCI does not mean silencing the entire X chromosome during meiosis, this is also not what we or other researchers have observed. As among about the 2000 genes that are located on the X chromosome, there must be some genes that are required during meiosis, and they cannot be silenced. This has already been clearly demonstrated in our Figure 1c, that although there is a significant underrepresentation of meiotic genes on the X chromosome, but we never stated that there are NO meiotic genes on the X, or all the X-linked genes have been completely silenced. Hence it should not appear as a surprise for the reviewer that there are ‘genes that stay on the X and avoid silencing’. As long as there are significant differences between the X chromosome and the autosome, which was also the logic of all previous studies (as we cited ref. 11, 12, 23 and 24), that indicates the action of MSCI.
